# Fantastic Ferulic Acid Esterases and Their Functions

**DOI:** 10.3390/ijms26157474

**Published:** 2025-08-02

**Authors:** Savvina Leontakianakou, Patrick Adlercreutz, Eva Nordberg Karlsson

**Affiliations:** Division of Biotechnology and Applied Microbiology, Department of Process and Life Science Engineering, Lund University, P.O. Box 124, SE-22100 Lund, Sweden; savvina.leontakianakou@ple.lth.se (S.L.); patrick.adlercreutz@ple.lth.se (P.A.)

**Keywords:** hydroxycinnamic acids, catalysis, bioconversion

## Abstract

Ferulic acid (FA) is one of the most abundant hydroxycinnamic acids found in plant cell walls. Its dehydrodimers play an important role in maintaining the structural rigidity of the plant cell wall. Ferulic acid esterases (FAEs) act as debranching enzymes, cleaving the ester bond between FA and the substituted carbohydrate moieties in FA-containing polysaccharides in the plant cell wall. This enzymatic reaction facilitates the degradation of lignocellulosic materials and is crucial for the efficient utilization of biomass resources. This review focuses on the occurrence of ferulic acid in nature and its different forms and outlines the various classification systems of FAEs, their substrate specificity, and the synergistic interactions of these enzymes with other CAZymes. Additionally, it highlights the various methods that have been developed for detecting hydroxycinnamic acids and estimating the enzyme activity, as well as the versatile applications of ferulic acid.

## 1. Introduction

Agricultural residues derived from plant cell walls can be converted to fermentable sugars that can be used in production of biofuels and chemicals. Today’s concern with developing alternatives from biobased sources has enhanced the interest in such materials and increased the amount of research in that field [1]. Lignin and hemicellulose together account for 40–70% of plant biomass by weight [2]. However, the complete degradation of lignocellulosic material is challenging due to its complex composition and the strong interactions between its components, which make it resistant to hydrolysis. Ferulic acid (FA) further contributes to this recalcitrance by acting as a structural component, substituting carbohydrate moieties in the xylan chain, the primary constituent of hemicellulose.

Ferulic acid, or its IUPAC name 3-methoxy-4-hydroxycinnamic acid, is covalently bound to polysaccharides such as xylan, galactoglucomannan, and pectin. It plays a crucial role in cross-linking polysaccharide chains and linking hemicellulose to lignin through radical coupling with monolignols in ferulated xylan [2].

Feruloyl esterases (E.C. 3.1.1.73), also known as ferulic acid esterases (FAEs), cinnamoyl esterases, cinnamoyl ester hydrolases, and *p*-coumaroyl esterases, belong to the carboxyl esterase family. These enzymes break the bond between FA (or other hydroxycinnamic acids) and polysaccharides. They play a key role in biomass degradation by working synergistically with other enzymes to release sugar monomers. Due to their catalytic abilities, FAEs have widespread applications in industries such as pharmaceuticals, agriculture, and the biofuels industry.

## 2. Origin and Occurrence of Different Ferulic Acid Forms

FA belongs to the broader category of hydroxycinnamic acids, alongside *p*-coumaric acid, caffeic acid, and sinapic acid (see Figure 1). These compounds form the largest class of phenolic compounds, which are extensively studied for their antioxidant activity and their role in preventing diseases, including cancer, cardiovascular disorders, and neurodegenerative conditions [3,4].

Ferulic acid is often found linked to polysaccharides, primarily glucuronoarabinoxylan (GAX) and, more frequently, arabinoxylan (AX), as glucuronic acid is not widely found in cereal grains. GAX consists of β-(1→4)-linked xylose units that are branched via glycosidic bonds to glucuronic or methyl glucuronic acid at the C(O)2 position, and α-L-arabinosyl at the C(O)2 and/or C(O)3 positions. In AX, glucuronic acid is absent, and the structure primarily comprises β-(1→4)-linked xylopyranosyl units with α-L-arabinosyl substitutions at the C(O)2 and/or C(O)3 positions.

In monocotyledons, particularly in the Poaceae family (e.g., wheat, maize, barley, and rice), ferulic acid has been extensively studied [5]. Here, ferulic acid is linked through its carboxyl group to the C(O)5 of arabinose, forming 3-O-(5-O-trans-feruloyl-α-L-arabinofuranosyl)-(1→3)-D-xylose (Figure 2a). In bamboo shoots, ferulic acid connects to the xylose backbone, forming O-(4-O-trans-feruloyl-α-D-xylopyranosyl)-(1→6)-D-glucopyranose [6] (Figure 2b).

Aside from ferulic acid, *p*-coumaric acid is also present in plant cell walls in smaller amounts. Interestingly, in bamboo shoots, *p*-coumaric acid is linked to arabinose via an ester bond, forming 3-O-[5-O-(trans-*p*-coumaroyl)-α-L-arabinofuranosyl]-(1→3)-O-β-D-xylopyranosyl-(1→4)-D-xylopyranose (Figure 2c). Both ferulic and *p*-coumaric acids exist predominantly in their trans-isomeric forms. Cis-ferulic acid has been reported in significantly smaller amounts, comprising approximately 2–18% of the total ferulic acid content in maize roots [7]. However, it remains understudied, primarily due to challenges in separating the cis and trans isomers [8].

In dicotyledons, ferulic acid is found in smaller amounts compared to monocotyledons. It serves as a side chain of arabinan and galactan in sugar beet pulp [9] and spinach. The latter can also be esterified with a *p*-coumaric acid in the primary cell wall [10]. In galactan, ferulic acid forms an ester bond at the C(O)6 position, while in arabinan, it forms ester bonds at the C(O)2 or C(O)3 positions [11]. Evidence of its presence have been found in Chinese water chestnut, carrots, glass wort, and pine hypocotyl cell walls [12,13,14,15].

## 3. Ferulate Dimers and Cross-Links

Ferulic acid has a bifunctional role in plant cell walls. Ferulate can form dimers either through the aromatic ring or the aliphatic chain. These dimers result in cross-links between polysaccharide–polysaccharide chains or polysaccharide–lignin chains, influencing the accessibility and digestibility of the cell wall [16].

The general structure of ferulate consists of the ferulic acid aromatic ring which is linked via an ether bond to lignin, while the carboxyl group forms an ester bond with the polysaccharide chain [17]. Interestingly, ferulates act as initiation sites for the lignification process through direct covalent binding to monolignols such as coniferyl and sinapyl alcohols [18].

The formation of ferulate dimers in polysaccharides can occur through two pathways: photochemically [19] and via peroxidase-catalyzed oxidative coupling. The first method produces cyclodimers [20,21], whereas radical-mediated dimerization results in a wide range of dehydrodimers [18].

The five main type of dehydrodimers (Figure 3) found in nature are [18]:8-O-4′-DiFA:(Z)-β-{4-[(E)-2-carboxyvinyl]-2-methoxyphenoxy}-4-hydroxy-3-methoxy-cinnamic acid;8-5′ DiFA: (E,E)-4,4′-dihydroxy-3,5′-dimethoxy-β,3′-bicinnamic acid;8-5′-BenDiFA: in the benzofuran form;8-8′-DiFA: (E,E)-4,4′-dihydroxy-3,3′-dimethoxy-β,β′-bicinnamic acid;5-5′ DiFA: (E,E)-4,4′-dihydroxy-5,5′dimethoxy-β,3′-bicinnamic acid.

The exact type of dehydrodimers in a sample can be determined after their isolation (using enzymatic degradation or chemical saponification) and depends on the species that are being treated. Diferulates are present in a wide range of cereals such as maize, wheat, spelt, rice, wild rice, barley, rye, oats, and millet. For example, Bartolome et al. reported that in alkali- or enzyme-treated spent barley grain, 5′-5′ DiFA was the predominant diferulate [22]. It is estimated that the diferulate content is 8–39 times higher in the insoluble dietary fiber fraction than in the soluble one. The proportions of various dimers are different in the two fractions, with 8-5′ DiFA being predominant in the insoluble fraction and 8-8′ DiFA in the soluble [23]. Hatfield reported that model substrates of feruloylated arabinose mainly resulted in formation of 8-5′ DiFA and 8-5′ DiFA benzofuran [18].

Oxidative coupling is not only limited to the formation of dehydrodimers. Several findings support the presence of both dehydrotrimers (Figure 4) and dehydrotetramers in maize bran [24,25,26,27,28,29]. These structures include the 5-5/8-O-4-coupled dehydrotrimer [27], which is hypothesized to be the prevailing trimer in corn bran, the 8-O-4/8-O-4-dehydrotrimer, and the 8-8 (cyclic)/8-O-4-dehydrotrimer [24]. The isolation of 8-O-4/8-O-4- and 8-8(cyclic)/8-O-4-coupled dehydrotrimers suggests that radical-coupled ferulate dehydrotrimers could potentially lead to cross-linking of three polysaccharide chains [24].

The two tetramers that have been isolated today are probably formed by 4-O-5 coupling of two DiFA 8-5′ (cyclic), and by 5-5′ DiFA−DiFA coupling [29].

Hydroxycinnamic acids were also widely hypothesized to covalently cross-link with Tyr and Cys residues in the cell wall proteins [30]. Proteins have multiple roles in the cell wall, such as transport, inhibitory functions, and structural purposes. Their secretion influences their solubility state, and these insoluble proteins are hypothesized to have ionic or salt interactions with polysaccharides [30]. Although no evidence that these linkages exist in nature has been reported yet, in vitro experiments have been conducted, and covalent coupling between proteins and polysaccharides is expected to result in new macromolecules with functional properties. FA could act as a substrate for the formation of FA oligomers in the presence of horseradish peroxidase (HRP) and H_2_O_2_, and this degree of oligomerization increases when the FA is bound to Gly-Tyr-Gly tripeptide [16,31]. In 2004, Boeriu et al. achieved cross-linking between protein and an FA polysaccharide. In the presence of HRP and H_2_O_2_, they enzymatically conjugated arabinoxylan to β-casein, which contains Tyr residues [32].

## 4. Feruloyl (Ferulic Acid) Esterases and Their Classification

FAEs are an important subclass of carboxylic esterases (E.C. 3.1.1), since they facilitate the degradation of plant cell walls via hydrolysis of the ester bond. The release of the ferulate ester groups can result in breaking of the cross-link between hemicellulose chains or between hemicellulose and lignin. FAEs usually exist in a monomeric form with a single catalytic site, and in some cases contain a carbohydrate-binding module (CBM). The presence of a CBM in FAEs has been shown to improve the catalytic efficiency of the protein [33,34,35,36,37,38]. Multiple attempts have been made to systematically classify these enzymes, and different systems have been developed over the years when trying to address their diversity. Below we summarize the main aspects, key features, and limitations of these systems (Table 1).

FAEs as carbohydrate-active enzymes are initially classified under the Carbohydrate Esterase Family 1 (CE1) in the CAZy database. This classification, with the criteria of sequence similarity and being within the same CE family, includes trehalose 6-O-mycolyltransferase (EC 2.3.1.122), diacylglycerol O-acyltransferase (EC 2.3.1.20) carboxylic ester hydrolase (EC 3.1.1), and acetyl xylan esterase (EC 3.1.1.72) [43]. Based on the CAZy database, additionally to CE1, one FAE is classified in CE15 as a 4-O-methyl-glucuronoyl esterase/feruloyl esterase (FI6_CE15).

A widely used FAE classification system is the one proposed by Crepin et al. in 2004 [39]. The criteria used for clustering were primary sequence identity and activity against synthetic substrates and diferulate. Based on these criteria, four subclasses were proposed: A, B, C, and D.

Four model substrates were used to estimate activity and were the methyl esters of the acids. These types of hydroxycinnamic acids are methyl ferulate (MFA), methyl caffeate (MCA), methyl sinapate (MSA), and methyl *p*-coumarate (M*p*CA). All enzymes from types C and D were active on all synthetic substrates, whereas type A was not active on MCA, and type B was not active on MSA. Additionally, type B and C enzymes were unable to release ferulic acid dehydrodimers from model and complex substrates [39].

A phylogenetic tree was constructed based on the available sequences, highlighting their similarity to other enzyme families. Type A enzymes resemble lipases, type B is similar to acetyl xylan esterases, type C is related to chlorogenate esterase and tannase, and type D shares similarities with xylanases. In addition, six sequences clustered outside this subclassification scheme, suggesting the existence of a fifth class, E.

The system, however, has certain limitations, which are acknowledged by the authors. These include a lack of comparable activity data, a reliance on primary sequence identity, and gaps in the knowledge regarding the type of biomass that triggers the gene expression of these proteins [39].

In 2008, the presence of FAEs in fungal genomes was extensively investigated by Benoit et al [40]. Starting with a strict threshold 1E*^−^*^70^ of BlastP and continuing with phylogenetic analysis, multiple alignment (MUSCLE), and tree reconstruction methods, they proposed seven FAE subfamilies. Biochemically characterized members were included in three of these subfamilies (SF). The biochemical properties of these enzymes appeared to be different, suggesting that the SF described different classes of FAEs [40]. However, the number of FAE genes varies significantly, even among closely related species. This suggests that FAEs evolved from different esterase families and do not necessarily have a common ancestor. Some of these diverse esterase families that FAEs descent from have been classified in specific SFs, such as SF1-4 from tannases, SF6 from acetyl xylan esterase, and SF7 from lipases. However, in comparison to Crepin’s system, Type D FAEs are missing in Benoit’s analysis.

In parallel, in 2011, Udatha et al. [41] conducted a descriptor-based computational analysis that led to a broader classification of FAEs from fungi, bacteria, and plants. This system categorized FAEs into 12 families, with further subclassification within each family to distinguish substrate specificities [41].

Benoit’s work on fungal genes was later expanded by Dilokpimol et al. in 2016 [42]. They investigated the evolutionary relationships between the fungal FAE families and other related enzyme families such as acetyl xylan esterases, lipases, and tannases. Their phylogenetic tree analysis resulted in 13 FAE subfamilies. Some subfamilies, such as SF1–SF3 and SF5–SF7, remained consistent with earlier classifications, while SF8–SF13 represented newly identified evolutionary groups, highlighting the diversity within FAEs [42]. This work managed to provide a complete classification of fungal FAEs and illustrated that despite the catalytic triad Ser-His-Asp being conserved, FAEs do not have a common ancestor.

Moreover, this updated phylogenetic analysis demonstrated that Crepin’s ABCD system does not align well with the evolutionary relationships of FAEs. Some subfamilies are predominantly associated with a single type (A, B, C, or D), while others include multiple types, indicating greater complexity than previously understood.

## 5. Catalytic Mechanism and Well-Characterized FAEs

The catalytic mechanism of FAEs has been widely studied and is similar to that adopted by lipases, following the mechanism of serine hydrolases (Figure 5). The catalytic triad consists of Ser-His-Asp/Glu, and the consensus sequence GXSXG is surrounding the Ser. In brief, the hydroxyl group of serine (nucleophile) attacks the carbonyl carbon of the ester. This results in the formation of a tetrahedral intermediate, which then collapses to generate an acyl-enzyme intermediate, with the acyl moiety being covalently bound to the enzyme.

In the second step, a water molecule acts as a nucleophile, hydrolyzing the acyl-enzyme intermediate. Along with the protonation of the ester oxygen by His, this leads to the release of the acid in the deacylation step. In both steps, tetrahedral intermediates are formed and are energetically stabilized by the oxyanion hole, where two NH groups form hydrogen bonds with the carbonyl oxygen [44]. Specifically, for FAEs, the catalytic acid can interact directly with the para-hydroxy group, forming a hydrogen bond. Upon mutation, the esterase activity is maintained, but the recognition on hydroxycinnamate acids is abolished [45]. In some cases the same hydrogen bond is hypothesized to restrict the activity only on monomeric FA and prevent the recognition of dimers [46].

Currently, in the BRENDA database (brenda-enzyme.info) on functional and molecular information of enzymes, only 13 PDB entries and structure links are available, and all these enzymes have an α/β fold. Only few detailed studies on FAEs are available and originate from *Aspergillus niger* [47,48,49,50,51,52] and *Clostridium thermocellum* [53,54,55] (Figure 6). For example, *An*FAE from *A. niger* is a α/β hydrolase, with the catalytic triad being described above [48,49,51]. This protein has a significant sequence similarity of 37% to a lipase from *Rhizomucor miehi* but does not exhibit any lipase activity [51,52]. Another detailed study on FAEs concerns enzymes designated XynY and XynZ [53,54,55], originating from *Clostridium thermocellum*. They exhibit a canonical α/β-hydrolase fold based on an eight-stranded β-sheet surrounded by α-helices. Despite this, they exhibit a poor structural homology with *An*FAE, although the catalytic site of all three enzymes is well conserved.

A number of different FAEs are characterized to varying degrees (see Table 2) and occur either as monomers or dimers. Some FAEs have interesting and distinct features, including activity in the presence of various metal ions or activity under more extreme conditions (pH and/or temperature).

A metal ion-tolerant, 55kDa monomeric FAE from the mushroom *Lactarius hatsudake* (LhFAE) was characterized in 2016. LhFAE maintained 80% of the initial enzyme activity in the 5.0 mM presence of K^+^, Al^3+^, Pb^2+^, Cd^2+^, Mn^2+^, Mg^2+^, Zn^2+^, and Ca^2+^ ions [59].

Stability at extreme pH values has also been targeted. In 2005, Topakas et al. reported an FAE, designated StFaeC, from *Sporotrichum thermophile*, which was active on all methyl hydroxycinnamic acids and classified as type C. StFaeC was a dimer in solution and had an optimal pH at 6.0, but still retained 50% of its activity at pH 10 [57]. A low-pH-tolerant FAE, PpFAE, was isolated from *Penicillium piceum. Pp*FAE exhibited activity at a broad pH range of 2.0–8.0 with an optimum at 3.0, while the optimum temperature was 70 °C [58]. As most FAEs exhibit optima under less harsh conditions (often pH 6–7 and temperature up to 50 °C), the high acidic thermal stability suggests wider application fields for this enzyme.

Further work includes the engineering of FAEs. In 2022, Yang et al. reported a new FAE from *Geobacillus thermoglucosidasius* and performed site-directed mutagenesis to enhance thermostability. Although the mutations increased the T_m_, the enzyme activity decreased for all variants. The most favorable outcome (highest T_m_) was observed in a double mutant. In this variant, an arginine substitution led to salt bridge formation and reduced flexibility in the cap region, while an asparagine substitution improved the hydrophilicity of the protein surface [61]. This indicates that the flexibility of the catalytic region is of importance for the activity, which was corroborated for a novel FAE from *Lactobacillus acidophilus,* designated as LaFae [62]. In this dimeric protein, a phenylalanine-to-alanine mutation near the catalytic site increased the activity. This variant probably exhibits enhanced flexibility of the cap domain, facilitating substrate binding [62].

Accessibility to the catalytic site may be another factor of importance. In 2022, Kasmaei et al. reported a new type of FAE homodimer in *Lb*FAE from *Lentilactobacillus buchneri.* This enzyme exhibits a unique dimerization profile, and the active site of each monomer is located internally, facing the other protomer. This enzyme can only accommodate a single FA and is not active on dehydrodimers [45]. This contrasts with the findings of Lai et al. (2011), in whose study the characterized dimeric protein from *Lactobacillus johnsonii* N6.2 (LJ0536) demonstrated a solvent-exposed, open canal, large catalytic site [60].

These diverse structural and functional studies emphasize the versatility of FAEs across different microbial sources. The number of structurally resolved enzymes is still limited. However recent advances in protein engineering and characterization highlight FAEs’ potential for industrial and biotechnological applications.

## 6. FAEs in (Trans)Esterification

Various studies have demonstrated the versatility of fungal FAEs in a wide variety of reaction media containing mainly non-aqueous solvents, enabling efficient esterification and transesterification of phenolic acids with simple alcohols and with carbohydrates. For some applications, free ferulic acid is the most useful substance, but in other cases, derivatives of ferulic acid have more promising properties. Therefore, derivatization of ferulic acid, for example by esterification, is of great interest. Some of these cases are highlighted in the following section.

Giuliani et al. used an FAE to esterify ferulic acid with 1-pentanol in a microemulsion containing n-hexane, water, and the surfactant cetyltrimethylammoniumbromide [68]. A high yield was obtained (up to 60%), but due to the low solubility of ferulic acid in the system, the reactions were carried out at concentrations below 1 mM.

In 2003, Topakas et al. purified and characterized an FAE-II from *Fusarium oxysporum* that retained its catalytic activity in a surfactantless microemulsion containing n-hexane, 1-propanol, and water, in which the solubility of cinnamic acid derivatives was up to 100 times higher. This type of microemulsion is also called a ternary solvent system and has been used in several subsequent studies of FAE-catalyzed synthesis reactions. The *F. oxysporum* enzyme catalyzed the esterification of various phenolic acids but not cinnamic acid, indicating the necessity of a C-4 substitution on the phenyl ring [64].

In 2005, the same group reported the first enzymatic feruloylation of a carbohydrate, L-arabinose, with *Sporotrichum thermophile* FAE type C (StFaeC), in another ternary system consisting of n-hexane/t-butanol/water [57]. Other ternary systems were used for the feruloylation of di- and oligo-saccharides (*n*-hexane, 2-butanone, and MES–NaOH buffer), using an FAE from a commercial multi-enzyme preparation as a catalyst [69]. This composition was slightly altered for the synthesis of feruloylated galactobiose, where 2-butanone was exchanged with 1,4-dioxane [69].

Optimization of butyl hydroxycinnamate esters by transesterification of methyl esters catalyzed by a novel type C FAE from *Aspergillus ochraceus* was carried out in a ternary system containing isooctane/butanol/water [65]. This enzyme was quite specific for butanol, while other alcohols were converted at much lower rates. Synthesis of feruloyl glycerol esters was achieved in a solvent-free system of just ferulic acid in a mixture of glycerol and water using re-AoFaeA as a catalyst, achieving a yield of 60.3% [70].

Successful FAE-catalyzed transesterification reactions in aqueous/organic two-phase systems were carried out using *Talaromyces wortmannii* FAEs as catalysts [66]. In the reaction between vinyl ferulate and prenol, just n-hexane with a small addition of water was used as the reaction medium, and in the reaction between vinyl ferulate and L-arabinose, a small amount of DMSO was added as well. The reactions were optimized, giving yields of 87.5% for prenol and 56.2% for L-arabinose. In the case of prenyl ferulate synthesis, the aqueous phase was recovered after the reaction, making it possible to reuse the enzyme, a procedure which was repeated for six consecutive cycles.

## 7. Synergistic Degradation and Bifunctional Esterases

Given the complexity of the plant cell wall, it is natural to assume that different types of enzymes are necessary for its efficient degradation. The increased activity of enzymes when combined with others from different classes is referred to as *synergy*. There are multiple reports of FAEs working synergistically with various carbohydrate-degrading enzymes to break down different biomass side streams. FAEs have demonstrated synergistic effects with xylanases [47,57,59,70], cellulases [58], and laccases. The combination of AnXyn11A and AnFaeA from *Aspergillus niger* BE-2 increased the initial release of ferulic acid (FA) from destarched wheat bran from 16.8% to 61% when added simultaneously [71]. In addition, an engineered bifunctional xylanase/feruloyl esterase (XynII-Fae) from *Prevotella ruminicola* 23 showed synergistic effects with commercial cellulase for the delignification of corn stover [67]. In this engineered bifunctional protein, each reaction was catalyzed by separate domains, with a TIM barrel and α/β hydrolase folding, respectively. A recent study from Schmitz et al., conducted in 2022 [72], examined the degree of synergy for multiple commercial carbohydrate enzymes in different biomass types, including corn bran, soluble oat bran, insoluble oat bran, and oat hull. The study found varying levels of synergy depending on the substrate [72]. In this work, FAEs showed synergy with laccase and xylanases from GH families 5, 10, and 11. However, not all side streams exhibited this effect, suggesting that the composition and nature of the biomass play a determining role in the degree of degradation [72].

The chemical complexity of plant biomass has directed evolution towards enzymes with dual functionality. This dual activity can occur within the same type of bond, such as esters, or span across different bond types, like ester and glycosidic bonds. For example, Dilokpimol et al. (2022) reported three novel feruloyl/acetyl xylan esterases (FXEs) that are capable of efficiently releasing both ferulic acid and acetic acid from wheat arabinoxylan [73]. Earlier, in 2000, Blum et al. demonstrated that XynY and XynZ, components of the *Clostridium thermocellum* cellulosome, exhibited FAE activity originating from their C-terminal and N-terminal domains, respectively, in addition to their established xylanase activity. The xylanase function in these proteins is associated with GH family 10, while the feruloyl activity originates from distinct catalytic domains [55]. Here again, the enzyme is not a single fold carrying out both activities, it is a multi-domain structure with dual functionality. More recently, Schmitz et al. (2023) reported a bifunctional arabinoxylanase/feruloyl esterase belonging to GH family 5, in which both activities share the same active site. In that case, the recognition of the arabinose substitution is specifically required for xylanase activity [74].

## 8. Estimation of Enzymatic Activity and Detection of FA

The catalytic activity of FAEs can be studied using either complex natural substrates or synthetic model compounds. When complex substrates are used, the released FA is monitored. High-Performance Liquid Chromatography (HPLC) with a C-18 reverse-phase column is commonly employed for quantitative analysis, typically using UV detection at 320 nm [75]. Gas Chromatography–Mass Spectrometry (GC-MS) is also suitable for the detection, quantification, and separation of geometrical isomers of FA [76]. Kvasnička et al. separated 12 phenolic acids including FA, with capillary zone electrophoresis (CZE) using HPLC with UV detection [77]. There are multiply reports of Liquid Chromatography–Tandem Mass Spectrometry (LC-MS/MS) being used for the identification and quantification of FA in a diverse range of substrates, such as Chinese herbal medicine [78], rat plasma [79], and milk [80].

Qualitative detection can be achieved with thin-layer chromatography TLC [81] and high-performance thin-layer chromatography (HPTLC), which have been proven effective for the separation and detection of ferulic acid and vanillin [82].

In terms of synthetic substrates, methyl esters of hydroxycinnamic acids are frequently used, particularly for enzyme classification into types A, B, C, and D. Methyl ferulate (MFA), methyl caffeate (MCA), methyl sinapate (MSA), and methyl *p*-coumarate (M*p*CA) are incubated with the enzyme, and the hydrolysis products are analyzed via HPLC, providing a detailed estimation of the enzyme activity with these different substrates [39,48,83]. Another widely used method involves the use of *p*-nitrophenyl (*p*NP) esters, such as *p*NP-ferulate or *p*NP-acetate, in spectrophotometric assays [83]. These allow for real-time monitoring of enzyme activity by measuring the release of *p*-nitrophenol at 410 nm.

Frequently, FAEs exhibit dual activity as feruloyl esterases and acetyl xylan esterases. It is therefore of interest to test the enzymatic activity and preference in both *p*NP-ferulate and *p*NP-acetate [73,74,84]. These synthetic substrate assays are advantageous for enzyme screenings, since they are compatible with high-throughput methods and kinetic analysis and can facilitate FAE classification. Nevertheless, it is important to note that they do not reflect the complexity found in nature. Therefore, high activity on synthetic substrates does not necessarily translate to efficient activity on plant cell wall materials.

## 9. Applications of FAEs and FA

FAEs’ applications can be viewed in two ways. One focuses on their enzymatic effect on biomass, where degrading the side chain enhances the accessibility of hydrolytic enzymes to cellulosic and hemicellulosic substrates and lignin–polysaccharide chains (described in the Section 7). The other considers FA as a valuable product with diverse applications in pharmaceuticals, cosmetics, food, feed, food packaging, and biogels [85].

In pharmaceuticals and cosmetics, FA is widely used for its antioxidant, anti-inflammatory, anti-aging, and photoprotective properties [85,86]. It is an important ingredient in sunscreens and skin-whitening products due to its ability to inhibit melanin accumulation, protect against UV radiation, and prevent skin photoaging [86,87,88].

Additionally, FA exhibits anti-cancer properties by inhibiting reactive oxygen species (ROS), protecting cellular components from oxidative damage, and regulating intracellular signaling related to proliferation, apoptosis, and metastasis. FA and its derivatives also offer antithrombotic, immune-regulating, and neuroprotective benefits, aiding in the treatment of cardiovascular and cerebrovascular diseases [89]. Studies further suggest wound-healing properties, demonstrated in experiments on diabetic rats [90,91]. These benefits derive from FA’s free radical scavenging ability and reduction in oxidative damage [85]. FA can be used as substrate in enzymatic decarboxylation reactions producing styrene derivatives, which can be converted further to various products, such as biobased polymers [92,93,94].

In the food industry, FA is a precursor for vanillin production [95], a flavoring compound that is also used in drug delivery systems and as battery energy storage through vanillin-cross-linked chitosan electrodes [96].

FA is also incorporated into fibrous films for fruit preservation, where its core–sheath-structured fibers provide both immediate and sustained antioxidant release, helping to prevent oxidation, reduce water loss, and extend shelf life [97]. Despite its broad potential across various fields, FA’s practical applications remain limited due to its low water solubility [85].

## 10. Conclusions

This review aims to provide a comprehensive overview of FAEs and ferulic acid (FA). It explores the various natural forms and cross-links of ferulic acid, emphasizing the compound’s structural complexity. All FAEs are serine hydrolases, playing an important role in the effective valorization of biomass, alone or in synergy with other enzymes. The released ferulic acid has broad applications across the pharmaceutical, food, and chemical industries. FAEs are also used as biosynthetic catalysts in (trans)esterification reactions involving non-aqueous solvents. By evaluating the current classification systems and the structural diversity of these enzymes, this review highlights the need for further research to fully elucidate their structure–function relationships and to deepen our understanding of the complex patterns of nature to which they correspond.

## Figures and Tables

**Figure 1 ijms-26-07474-f001:**
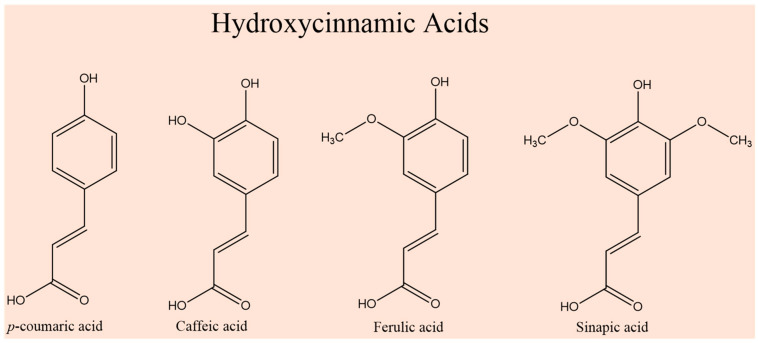
Illustration of the four main hydroxycinnamic acids, *p*-coumaric acid, caffeic acid, ferulic acid, and sinapic acid, highlighting their structural similarities and different functional groups on the phenyl ring.

**Figure 2 ijms-26-07474-f002:**
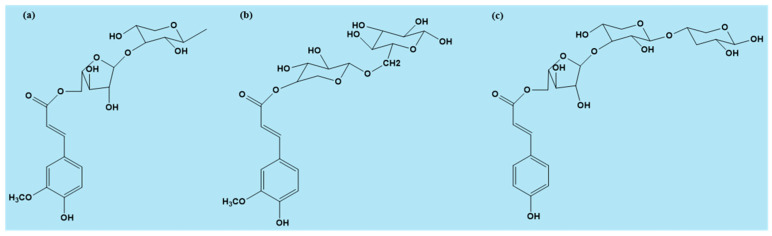
Hydroxycinnamic acids bound to a polysaccharide chain: (**a**) ferulic acid (FA) attached to arabinose, which is linked to xylose; (**b**) FA directly bound to xylose, which is connected to glucose; and (**c**) *p*-Coumaric acid attached to arabinose, which is linked to xylose.

**Figure 3 ijms-26-07474-f003:**
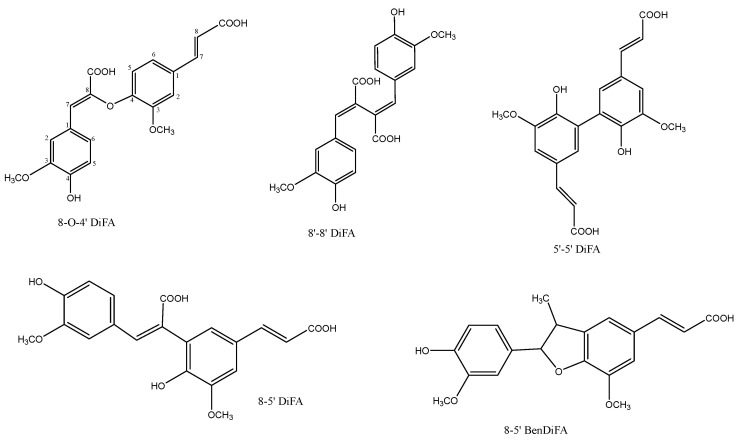
Structures of FA dehydrodimers.

**Figure 4 ijms-26-07474-f004:**
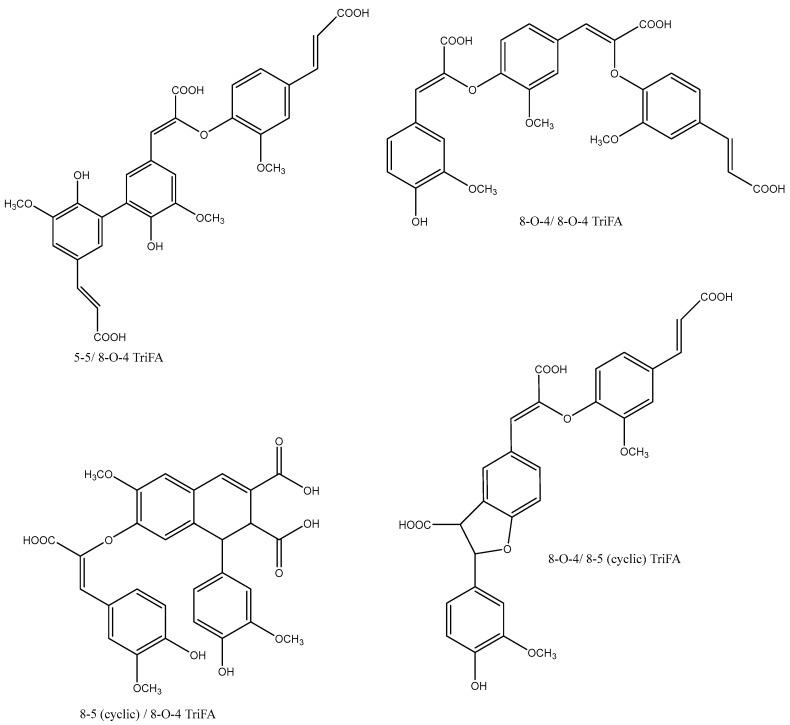
Structures of FA trimers.

**Figure 5 ijms-26-07474-f005:**
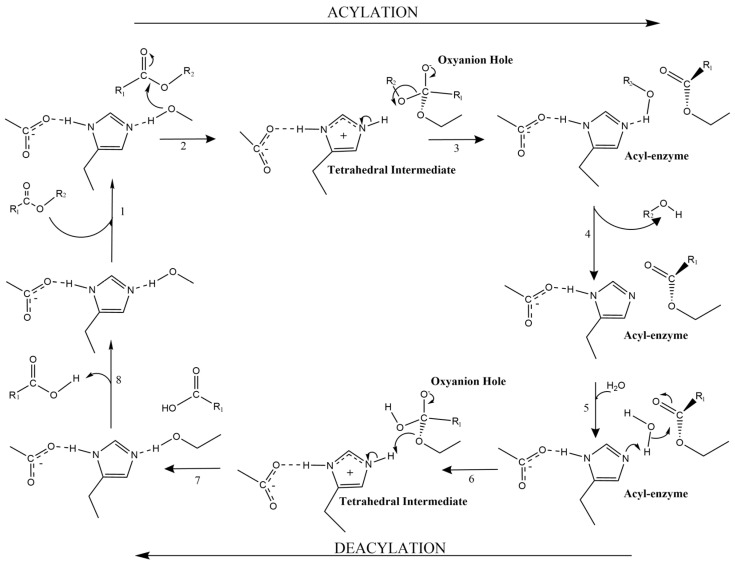
Schematic view of the serine hydrolases mechanism. The figure illustrates all the steps for carbohydrate hydrolysis and the formation of a tetrahedral intermediate, which is stabilized by the oxyanion hole and then collapses from the formation of the acyl enzyme complex during acylation. The addition of water initiates the reverse process of deacylation.

**Figure 6 ijms-26-07474-f006:**
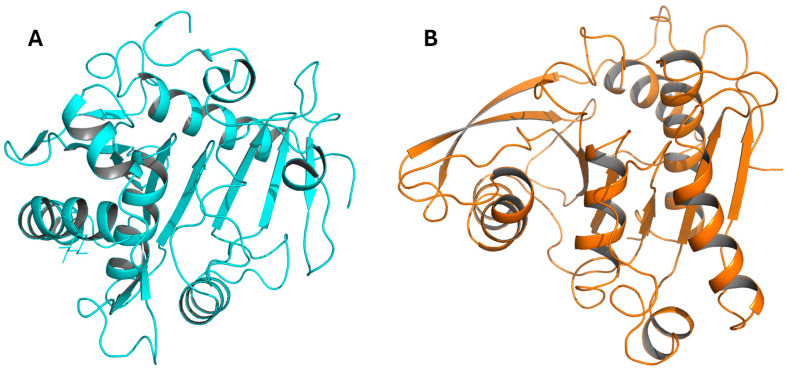
Cartoon diagrams of AnFAE, PDB 1UZA (**A**), and XynZ, PDB 1JJF (**B**), highlighting the low overall structural homology between the two enzymes, which are both classified as FAEs.

**Table 1 ijms-26-07474-t001:** FAE classification systems.

Classification System	Description	Main Features	Limitations
CAZy (CE1)	Groups FAEs into the CE1 based on sequence similarity and enzymatic catalytic mechanism.	Standardized database for carbohydrate-active enzymes.	Does not consider the functional diversity of FAEs.
Crepin (2004) [39]	Categorizes FAEs into types A, B, C, D, and putative E based on sequence similarity and substrate specificity.	Correlations based on sequence similarities and specificity on synthetic methyl esters.	Based on limited data; it can cluster unrelated enzymes within the same group.
Benoit (2008) [40]	Seven fungal FAE subfamilies based on phylogenetic analysis of fungal genomes.	Identifies evolutionary relationships within fungal species.	Limited to fungal genomic data. Excludes bacterial FAEs.
Udatha (2011) [41]	Twelve FAE families based on sequence and structure from fungi, bacteria, and plants.	Validation via computational and experimental methods.	Lacks functional correlation for some enzymes.
Dilokpimol (2016) [42]	Thirteen subfamilies of fungal FAEs based on phylogenetics and substrate specificity.	Divides fungal FAEs into subfamilies within CE1 based on biochemical data from new fungal genomes.	Error-prone in grouping bacterial and fungal FAEs together.

**Table 2 ijms-26-07474-t002:** Summary of well-characterized FAEs.

Enzyme	Organism	Oligomeric State	pH Opt.	Temp Opt.	Interesting Characteristics
AnFAEA	*Aspergillus niger*	Monomer	6.0 *	37 °C *	Tyr80 and Trp260 mutations to accommodate bigger substrates [46] (PDB 2BJH).
AoFAEB	*Aspergillus oryzae*	Dimer	7.0 *	37 °C *	C residues neighboring the catalytic S and H form a disulfide bond (motif found in tannases), which is crucial for activity [56] (PDB 3WMT).
XynZ	*Clostridium thermocellum*	Not mentioned	6.0	60 °C	CBM containing FAEs that are active with ΔCBM truncation but inactive when unknown (non-catalytic domain) is omitted [55].
StFaeC	*Sporotrichum thermophile*	Dimer	6.0	55 °C	Type C FAE, active on all methyl hydroxycinnamic acids [57].
PpFAE	*Penicillium piceum*	Not mentioned	3.0	70 °C	FAE type C with broad pH activity range 2.0–8.0 [58].
LhFAE	*Lactarius hatsudake*	Monomer	4.0	30 °C	Metal ion-tolerant: retains 80% of activity in 5 mM metal ions; Mn^2+^ boosts activity to 114% [59].
LbFAE	*Lentilactobacillus buchneri*	Dimer	6.5	40 °C	Internally facing active site; inactive on dehydrodimers [45].
LJ0536	*Lactobacillus johnsonii* N6.2	Dimer	-	-	Open canal catalytic site; solvent exposed [60].
GthFAE	*Geobacillus thermoglucosidasius*	Dimer	8.5	50 °C	Mutagenesis increased Tm, but activity decreased in variants [61].
LaFae	*Lactobacillus acidophilus*	Dimer	8.0	25–37 °C	Phe→Ala mutation near catalytic site increased activity; cap domain flexibility [62].
AmCE1	*Anaeromyces mucronatus*	Monomer	7.2 *	25 °C *	Structurally based loop domain “β-clamp” responsible for exolytic activity [63] (PDB 5CXU).
FAE-II	*Fusarium oxysporum*	Not mentioned	7.0	45 °C	Synergistic interaction with xylanase; active in a ternary solvent system [64].
AocFaeC	*Aspergillus ochraceus*	Monomer	6.5	40 °C	Butanol-specific biocatalyst, with 5x higher butyl caffeate synthesis rate compared to type B FAE from *A. niger* [65].
Fae125	*Talaromyces wortmannii*	Not mentioned	4.7/4.7	24.5 °C/38.9 °C	The optimal values of pH and temperature correspond to the synthesis of PFA and AFA, respectively [66] **.
XynII-Fae	*Prevotella ruminicola 23*	Not mentioned	7.0	40 °C	Bifunctional xylanase/feruloyl esterase, with each activity originating from a distinct domain [67].

* Optimizing conditions in these cases were not part of the published work, and the values mentioned in the table correspond to the ones used to determine the enzymatic activity. ** PFA: prenyl ferulate; AFA: L-arabinose ferulate.

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
