# Peer review of "Fantastic Ferulic Acid Esterases and Their Functions"

_ijms, 2025, doi:10.3390/ijms26157474_

Round 1

Reviewer 1 Report

Comments and Suggestions for Authors

Manuscript presents very interesting, thorough and comprehensive review focused on enzyme with enormous and increasing significance for concept of biorefinery and tendency to exploit waste and by-products of agri- and food industry. Manuscript covers not only information about enzyme Feruloyl  Esterases, but also covers detailed description of structures and linkages within biomass that include ferulic acid residues. In my opinion, manuscript can be published in current form, but it could also be valuable for future readers if authors could add additional info in Section 3 about enzymes that could catalyze degradation of ferulic dimers, trimers and tetramers.

Author Response

Comment: Manuscript presents very interesting, thorough and comprehensive review focused on enzyme with enormous and increasing significance for concept of biorefinery and tendency to exploit waste and by-products of agri- and food industry. Manuscript covers not only information about enzyme Feruloyl  Esterases, but also covers detailed description of structures and linkages within biomass that include ferulic acid residues. In my opinion, manuscript can be published in current form, but it could also be valuable for future readers if authors could add additional info in Section 3 about enzymes that could catalyze degradation of ferulic dimers, trimers and tetramers.

Response: We thank the reviewer for the kind feedback. Regarding enzymes acting on dimers or trimers, we were unable to provide a dedicated section due to limitations in available data in the literature. In most cases, enzyme activity has been assessed using model substrates. However, in Section 5, we do reference specific enzymes that have been experimentally shown to act on dehydrodimers and compare their activity with enzymes characterized as active only on monomeric ferulic acid.

Reviewer 2 Report

Comments and Suggestions for Authors

Some parts (such as enzyme classification and structural diversity) contain redundant material.  Consider condensing any overlapping content.

 The terms "FAE," "feruloyl esterase," and "ferulic acid esterase" are interchangeable.  A consistent naming convention would increase readability.

 Some figure captions are repeated or misnumbered (for example, Figures 2b and 2c appear twice).

 The section on dehydrodimers and trimers might benefit from more detailed chemical illustrations or simplified explanations.

The section on catalytic mechanisms is well-written, although it may benefit from a simplified graphic for easier understanding.

 Some sentences are lengthy and might be broken up for better reading.

 Include a graphical abstract or summary figure.

 Include a section on future views to explain emerging trends or research gaps.

 Feruloyl esterases (E.C. 3.1.1.73), also known as... p-coumaroyl esterases..." (p. 1).  Clarify that some FAEs can operate on p-coumaroyl esters, but the words are not interchangeable.

 In the article's abstract: "Ferulic acid... is the most abundant hydroxycinnamic acid found in plant cell walls." .  Rephrase as "one of the most abundant".

"FAEs are initially classified under CE1 in the CAZy database..." _ (p. 6).  Recognize that FAEs occur in various CE families, not just CE1.

 "FA = Ferulic Ecid"_ (p. 15).  Please correct to "Ferulic Acid."

Author Response

Thank you for the constructive comments! They were of great help to us. Detailed responses to each comment are listed below.

  1. Comment: Some parts (such as enzyme classification and structural diversity) contain redundant material.  Consider condensing any overlapping content.

Response: The text has been reviewed and condensed when applicable. If there is a specific part that should be corrected, we will be happy to do so.

  1. Comment: The terms "FAE," "feruloyl esterase," and "ferulic acid esterase" are interchangeable.  A consistent naming convention would increase readability.

Response: The enzyme name is now stream-lined as FAE when applicable to increase readability.

  1. Comment: Some figure captions are repeated or misnumbered (for example, Figures 2b and 2c appear twice).
  2. Response: Captions are now corrected.
  3. Comment: The section on dehydrodimers and trimers might benefit from more detailed chemical illustrations or simplified explanations.

Response: Figure 3 and 4 have been redrawn and hopefully provide a better understanding of the structures.

  1. Comment: The section on catalytic mechanisms is well-written, although it may benefit from a simplified graphic for easier understanding.

Response: The mechanism figure was adjusted. The reaction is now illustrated with an ester substrate, providing a simpler view.

  1. Some sentences are lengthy and might be broken up for better reading.

Response: Sentences were split to enhance readability and understanding

  1. Comment: Include a graphical abstract or summary figure.

Response: A graphical summary has been added.

  1. Comment: Include a section on future views to explain emerging trends or research gaps.

Response: We describe the emerging trends in different parts of the manuscript. For example, in section 5, well characterised FAEs are described along with the engineering done to enhance their industrial application. Also, in section 9 the FA applications are highlighted. Additionally, the research gaps are mentioned as a closing statement in the conclusions section.

  1. Comment: Feruloyl esterases (E.C. 3.1.1.73), also known as... p-coumaroyl esterases..." (p. 1).  Clarify that some FAEs can operate on p-coumaroyl esters, but the words are not interchangeable.

Response: Clarification was made on the next sentence of the paragraph

  1. Comment: In the article's abstract: "Ferulic acid... is the most abundant hydroxycinnamic acid found in plant cell walls." .  Rephrase as "one of the most abundant".

Response:  Text was adjusted according to reviewers’ recommendation.

  1. Comment: "FAEs are initially classified under CE1 in the CAZy database..." _ (p. 6).  Recognize that FAEs occur in various CE families, not just CE1.

Response:  A phrase is added acknowledging the occurrence of one FAE in CE15.

  1. Comment: "FA = Ferulic Ecid"_ (p. 15).  Please correct to "Ferulic Acid."

Response: Text was adjusted according to reviewers’ recommendation.

Reviewer 3 Report

Comments and Suggestions for Authors

The review provides an overview on esterase for the hydrolysis of ferulic acid esters. The review is comprehensive and covers relevant topics from the description of natural ferulic acid esters to application of the FA and experimental techniques. In particular, the part on the role in biodegradation and the synergy with other enzymes is very insightful. Publication after some revision is recommended,

The authors mention that FA and CA occur ‘predominantly’ as trans isomers. ‘predominantly‘ implies that also the cis isomer has found, this should be elaborated.

Carboxylic esterase belong to the alpha beta hydrolase fold enzymes. Do all described FAEs belong to this fold?

Quality of chemical structures: Figure 2: H atoms of methoxygroup should be subscript; Figure 3: The angles of the substituents on the aromatic rings and the trans-double bond are not correct. Sp2 C atoms have a 120 degree angle. Also, the pentacycle in 8-5’BenDiFa is very odd. Why are the atoms in one structure numbered, but not in the others? Figure 4: Some carboxylate groups are shown with all bonds, others at COOH. It would be better to show all in the same way. The angle and bond length of a methoxy group in 8-O-4/8-O-4 TriFa is not correct. Also,. The size of the chemical structures differs a lot. All chemical structures should be shown in the same layout, including same bond lengths, font size etc.

Figure 5: The way how the carboxylates are shown is not correct. It is ok to distribute the charge on the carboxylate on both O-atoms by using the solid/dashed lines, however, there are minus signs on both O’s, which is one too many. Usually, in such a case the minus is drawn between the O atoms. A carboxylate has only one charge, not two.

It is unclear to me why the authors drew the mechanism of hydrolysis of peptides as the paper deals with carboxyl ester hydrolysis. I recommend to modify the structure and show hydrolysis of a carboxyl ester instead.

The description of the mechanism is a bit generic. Are there any mechanistic features specific for feruloyl esterase? Is there any interaction known between active site-amino acids and the para-hydroxy group?

Figure 6 compares the structures of two ferulic acid esterases. Despite low sequence homology, both belong to the same fold, and should therefore be shown in the same orientation.

Section 6: The examples are very interesting. The motivation for the transesterification should be explained. What can be gained by binding FA to glycerol and other polyps and sugars?

Do the enzymes with dual functionality (eg. FA esters, xylanase activity) also to the alpha-beta hydrolase fold enzyme family?

In the applications of the products, also the decarboxylation by phenolic acid decarboxylase (10.1021/acssuschemeng.3c06513) and subsequent derivatization (vanillin synthesis 10.1155/2013/590359 acetylation 10.1038/s42004-024-01138-x, O-methylation 10.1002/cctc.202402027) should be mentioned.

Author Response

Thank you for the constructive comments! They were of great help to us. Detailed responses to each comment are listed below.

  1. Comment: The authors mention that FA and CA occur ‘predominantly’ as trans isomers. ‘predominantly‘ implies that also the cis isomer has found, this should be elaborated.

 Response: Indeed, FA exist in both isomers, and some comments regarding the cis form are now added in the main text.

  1. Comment: Carboxylic esterase belong to the alpha beta hydrolase fold enzymes. Do all described FAEs belong to this fold?

Response: Yes, all FAEs have α/β fold. This is now clarified in section 5.

  1. Comment: Quality of chemical structures: Figure 2: H atoms of methoxygroup should be subscript; Figure 3: The angles of the substituents on the aromatic rings and the trans-double bond are not correct. Sp2 C atoms have a 120 degree angle. Also, the pentacycle in 8-5’BenDiFa is very odd. Why are the atoms in one structure numbered, but not in the others? Figure 4: Some carboxylate groups are shown with all bonds, others at COOH. It would be better to show all in the same way. The angle and bond length of a methoxy group in 8-O-4/8-O-4 TriFa is not correct. Also,. The size of the chemical structures differs a lot. All chemical structures should be shown in the same layout, including same bond lengths, font size etc.

Response: In fig. 2 the Hs of the methoxy groups are now subscripted and fig 3 and 4 have been redrawn according to the suggestions.

  1. Comment: Figure 5: The way how the carboxylates are shown is not correct. It is ok to distribute the charge on the carboxylate on both O-atoms by using the solid/dashed lines, however, there are minus signs on both O’s, which is one too many. Usually, in such a case the minus is drawn between the O atoms. A carboxylate has only one charge, not two.

Response: We agree with the reviewer, and we have adjusted the figure accordingly.

  1. Comment: It is unclear to me why the authors drew the mechanism of hydrolysis of peptides as the paper deals with carboxyl ester hydrolysis. I recommend to modify the structure and show hydrolysis of a carboxyl ester instead.

Response: The figure was adjusted, and the reaction is now illustrated with an ester substrate.

  1. Comment: The description of the mechanism is a bit generic. Are there any mechanistic features specific for feruloyl esterase? Is there any interaction known between active site-amino acids and the para-hydroxy group?

Response: We would like to thank the reviewer for this insightful comment. Indeed, there is a reported recognition of the OH group in the para position that allows distinct specificity between hydroxycinnamic acids and cinnamic acids. This is now clarified in our text where the catalytic mechanism is described.

  1. Comment: Figure 6 compares the structures of two ferulic acid esterases. Despite low sequence homology, both belong to the same fold, and should therefore be shown in the same orientation.

Response: We agree and now fig 6 has been replaced. In the new figure the protein structures were first aligned, and each protein was coloured with a single colour to show both similarities and low homology.

  1. Comment: Section 6: The examples are very interesting. The motivation for the transesterification should be explained. What can be gained by binding FA to glycerol and other polyps and sugars?

Response: The introduction paragraph of section 6 was expanded motivating the transesterification of FA.

  1. Comment: Do the enzymes with dual functionality (eg. FA esters, xylanase activity) also to the alpha-beta hydrolase fold enzyme family?

Response: The bifunctional enzymes usually contain more than one domain with different folds. The esterase activity occurs from the α/β fold. Clarifications in section 7 have been made to explain this more carefully.

  1. Comment: In the applications of the products, also the decarboxylation by phenolic acid decarboxylase (10.1021/acssuschemeng.3c06513) and subsequent derivatization (vanillin synthesis 10.1155/2013/590359 acetylation 10.1038/s42004-024-01138-x, O-methylation 10.1002/cctc.202402027) should be mentioned.

Response: These suggestions are now incorporated in section 9 of the manuscript.